# The Identification of Risk Factors for Symptomatic Spinal Metastasis Onset: A Prospective Cohort Study of 128 Asymptomatic Spinal Metastasis Patients

**DOI:** 10.3390/cancers15041251

**Published:** 2023-02-16

**Authors:** Kenichiro Kakutani, Yutaro Kanda, Takashi Yurube, Yoshiki Takeoka, Kunihiko Miyazaki, Hiroki Ohnishi, Tomoya Matsuo, Masao Ryu, Kohei Kuroshima, Naotoshi Kumagai, Yoshiaki Hiranaka, Shinya Hayashi, Yuichi Hoshino, Hitomi Hara, Yoshitada Sakai, Ryosuke Kuroda

**Affiliations:** 1Department of Orthopaedic Surgery, Kobe University Graduate School of Medicine, 7-5-1 Kusunoki-cho, Chuo-ku, Kobe 650-0017, Japan; 2Division of Rehabilitation Medicine, Kobe University Graduate School of Medicine, 7-5-1 Kusunoki-cho, Chuo-ku, Kobe 650-0017, Japan

**Keywords:** symptomatic spinal metastases, risk factor, SINS, threshold, prospective cohort study

## Abstract

**Simple Summary:**

The risk factors for symptomatic spinal metastasis (SSM) onset remain unclear. This prospective cohort study aimed to statistically analyze the significant risk factors. One hundred and twenty-eight asymptomatic patients were prospectively registered. Data were collected from sixteen candidates regarding independent demographic and clinical factors, including Spinal Neoplastic Instability Score (SINS). Multivariate analysis was performed to identify the risk factors for SSM onset. Furthermore, the threshold was calculated from the receiver operating characteristic curve using the Youden index. Thirty-seven patients (28.9%) developed SSM during the follow-up period. The total SINS was identified as the most significant factor. The cut-off value for the SINS was 9.5 (sensitivity: 67.6%; specificity: 83.5%). This study identified the significant risk factors for SSM onset and the threshold of the SINS. If long-term survival is expected, patients with a SINS ≥ 10 should be considered for intervention to prevent SSM.

**Abstract:**

**Background:** Symptomatic spinal metastasis (SSM) decreases the activities of daily living (ADL) and quality of life of cancer patients. However, the risk factors for SSM onset remain unclear. This prospective cohort study aimed to statistically analyze the significant risk factors. **Methods:** From 2016 to 2018, 210 consecutive patients with spinal metastases were prospectively registered. Patients with SSM at the first consultation and those who were unable to be followed-up owing to poor general condition were excluded. The demographic factors (age, sex, primary cancer, performance status, and ADL), clinical factors (radiation therapy, chemotherapy, molecularly targeted drugs, and bone-modifying agents (BMAs)), and Spinal Neoplastic Instability Score (SINS) were evaluated. Multivariate analysis was performed to identify the risk factors for SSM onset. Furthermore, the threshold was calculated from the receiver operating characteristic curve using the Youden index. **Results:** Thirty-nine patients who presented with SSM at the first consultation and 43 patients who were unable to be followed-up owing to poor general condition were excluded. Finally, 128 asymptomatic patients were included. Thirty-seven patients (28.9%) developed SSM during the follow-up period. The total SINS (OR: 1.739; 95% CI: 1.345–2.250) was identified as the most significant factor. The cut-off value of the SINS was 9.5 (sensitivity: 67.6%; specificity: 83.5%). Twenty-five (62.5%) of the forty patients with a SINS ≥ 10 developed SSM within a mean of 5.5 months (95% CI: 1.17–9.83). Furthermore, all patients with a SINS ≥ 13 developed SSM (*n* = 5) within a mean of 1.37 months (95% CI: 0.0–3.01). **Conclusions:** This study identified the significant risk factors for SSM onset and the threshold of the SINS. If long-term survival is expected, patients with a SINS ≥ 10 should be considered for intervention to prevent SSM.

## 1. Introduction

The prevalence of cancer patients has been steadily increasing in the aged population, and the number of patients with bone/spinal metastasis has also increased. In 2019, there were an estimated 1.7 million new cancer cases in the USA [1]. Following the lungs and liver, the spinal part of the skeletal system is the third most frequent organ involved in metastasis [2]. Won et al. reported that spinal involvement might occur in up to 40% of cancer patients [3]. Spinal metastases usually progress asymptomatically until the terminal phase. However, 10–20% of those with spinal metastasis experience the destruction of supporting spinal elements and develop symptomatic spinal cord compression [4]. This symptomatic spinal metastasis (SSM), representing neurological dysfunction and intractable pain, causes a significant decrease in the performance status (PS), activities of daily living (ADL), and quality of life (QOL) of patients [5,6]. Consequently, patients with SSM often have to cancel their standard therapy for primary cancer, including chemotherapy, radiation therapy, and molecularly targeted drugs. In addition, the aim of treatment for metastatic disease is not only longer survival, but also better QOL. Obtaining a fine balance between survival and QOL is the ultimate goal of treatment for metastasis. Therefore, managing bone/spinal metastasis is essential. Although some reports have described the effectiveness of radiation therapy and spine surgery for SSM [7,8], once spinal metastasis develops and causes neurological dysfunction and intractable pain, unscheduled treatments are needed and impede the treatment schedule for primary cancer for several months. Therefore, the concept of bone management has been applied worldwide as a multidisciplinary approach to preventing skeletal-related events [9]. Furthermore, understanding the risk factors and natural course of spinal metastasis is critical for early diagnosis and the prevention of disease progression to symptomatic changes. However, few studies have assessed comprehensive cohorts regarding the natural history of SSM, and useful risk factors for the bone management of spinal metastases have not yet been established [10].

The most widely accepted method for classifying the mechanical stability of spinal metastases is the Spinal Instability Neoplastic Score (SINS) [11] proposed by the Spine Oncology Study Group. The SINS comprises six components to assess instability in the affected segments. Its interpretation is divided into three distinct categories according to the total score: stable (0–6), potentially unstable (7–12), and unstable (13–18). The SINS is valuable for deciding the use of spine surgery for spinal metastasis. However, the association between the SINS and symptomatic changes is still unclear. The specific objective of this prospective cohort study was to identify significant and useful risk factors for SSM based on a comprehensive evaluation, including the SINS.

## 2. Materials and Methods

### 2.1. Ethics Statement

This study was approved by the Institutional Review Board of our institution. Written informed consent was obtained from each patient for their participation in this study, in accordance with the World Medical Association Declaration of Helsinki and the laws and regulations of our country.

### 2.2. Patients

From 2016 to 2018, 210 consecutive patients with spinal metastases in a single institution were prospectively registered in this study. Spinal metastasis was diagnosed via plain radiography, computed tomography, and magnetic resonance imaging; however, when the diagnosis was hard, bone scintigraphy, positron emission tomography, and histological evaluations of needle biopsy samples were performed. We defined SSM as spinal metastases associated with progressive neurological deficits or intractable pain resistant to conservative care, including the use of opioids. Then, we monitored whether patients with spinal metastases experienced these symptomatic changes. Of these, we excluded patients who had already developed SSM at the first consultation at the orthopedic clinic and those who were unable to be followed-up. 

The risk factors for symptomatic changes were investigated. The independent factors for the ten candidates included the following: As demographic factors, age and gender were investigated. Malignancy was scored using the primary cancer category of the revised Tokuhashi score [12]. The Eastern Cooperative Oncology Group Performance Status (ECOGPS) [13] and Barthel index (BI) [14] were used to evaluate the health of patients at study enrollment. The total score of the SINS was used to assess the instability and characteristics of spinal involvement. If there were multiple lesions in a patient, the one with the highest score was included. As treatment-related factors, history of chemotherapy, radiotherapy, molecularly targeted drugs or hormonal therapy, and bone-modifying agents (BMAs) at study enrollment were investigated. In the current study, chemotherapy, hormonal therapy, and BMAs treated for more than 3 months were included. In addition to patients who had completed radiotherapy for metastatic spine tumors 1 month before the study enrollment, patients who were receiving radiotherapy at study enrollment were included in the patients with a history of radiotherapy. For the other regimens, the same definition was used.

The survival duration was defined as the time from the date of the first consultation to the date of the last follow-up or death. The families or transfer institutions of patients who died and could not consult our department after their last follow-up were contacted by telephone to obtain their information. Additionally, SSM-free survival duration was defined as the time from the date of the first visit to the date of SSM onset.

### 2.3. Statistical Analysis

All statistical analyses were performed using SPSS 13.0 (SPSS Inc., Chicago, IL, USA) with significance set at a *p*-value of <0.05. Parametric variables are expressed as means and ranges. Non-parametric variables are expressed as the median and interquartile range. The overall survival rate was calculated using the Kaplan–Meier method. Multivariate logistic regression analysis was performed to identify the association between demographic and clinical factors at study enrollment and the incidence of SSM. For a significant risk factor, the cut-off value was calculated from the receiver operating characteristic (ROC) curve using the Youden index. Additionally, Fisher’s exact test was used to compare the occurrence of SSM between patients with ≥the cut-off value and patients with <the cut-off value. In addition, the SSM-free survival rate was calculated using the Kaplan–Meier method.

## 3. Results

### 3.1. Demographic and Clinical Data

A total of 39 patients with SSM at the first consultation and 43 patients who were unable to be followed-up owing to poor general condition were excluded. Finally, 128 patients who were asymptomatic at the first consultation and able to be followed-up were enrolled in this study (Figure 1). The mean age at the first consultation was 68.7 (range: 32–90), and 85 of the 128 patients (66.4%) were ≥65 years. Demographic factors and clinical factors at study enrollment are shown in Table 1. 

The median PS and mean BI were PS2 and 79.7 (range 5–100), respectively, indicating independent daily life. The median survival time after the start of the study was 15.2 months (95% confidence interval (CI): 7.9–22.5) (Figure 2). The median observational duration was 8.5 months (interquartile range: 2.4–22.4). Lung cancer was the most common type of primary cancer. The other primary malignant tumors are listed in Table 2.

### 3.2. Risk Factors for SSM 

Thirty-seven patients (28.9%) developed SSM throughout the follow-up period, whereas 91 patients (71.1%) did not show symptomatic changes. The causes of SSM were neurological deficit due to spinal cord compression in 15 cases and intractable pian due to pathological fracture in 22 cases. The logistic regression analysis revealed total SINS (OR: 1.739; 95% CI: 1.345–2.250) as the most significant identified factor (Table 3).

Among these significant factors, the total SINS was identified as the only risk factor for the onset of SSM. To determine the cut-off value, the ROC curve of the total SINS at study registration was used (Figure 3). From the Youden index, the cut-off value was 9.5 (sensitivity: 67.6%; specificity: 83.5%). Next, the authors compared the patients with a SINS ≥ 10 and SINS ≤ 9. The number of patients with a SINS ≥ 10 was 40. Of these patients, 25 patients (62.5%) developed SSM. In contrast, twelve (13.6%) of the 88 patients with a SINS ≤ 9 exhibited SSM, and the difference was determined to be statistically significant using the log-rank test (*p* < 0.001; Figure 4). The median SSM-free survival time for patients with a SINS ≥ 10 was 5.5 months (95% CI: 1.17–9.83). 

The causes of SSM in patients with a SINS ≥ 10 were neurological deficit due to spinal cord compression in 12 cases and intractable pian due to pathological fracture in 13 cases, whereas in patients with a SINS < 9, these were the causes in 3 cases and 9 cases, respectively. 

In addition, patients with a SINS ≥ 13 were categorized as unstable, and a subgroup analysis was performed for these patients. All patients with a SINS ≥ 13 developed SSM (*n* = 5). Among the patients with a SINS of 10–12, the median SSM-free survival time from the start of the study was 6.93 months (95% CI: 3.29–10.6). In contrast, that of patients with a SINS ≥ 13 was 1.37 months (95% CI: 0.0–3.01), and the difference was significant (*p* = 0.001, Figure 5).

## 4. Discussion

Spinal metastasis is a growing global health problem in cancer patients. The treatment of spinal metastasis is aimed at maintaining and increasing the PS, ADL, and QOL and minimizing the adverse effects of primary cancer therapies. To achieve this, the early diagnosis and prevention of SSM are essential. Thus, the natural history of spinal metastases and useful risk factors for medical care must be revealed. However, the natural history of spinal metastases remains unclear. In the current study, the authors performed a prospective cohort study to determine the natural history of spinal metastasis and identify risk factors for symptomatic changes. This study revealed the significant risk factors and threshold for SSM onset. 

When considering the risk factors for SSM onset and intervention in patients with spinal metastases, mechanical instability is one of the most important factors. Currently, the SINS is widely used as a scoring system to assess the mechanical stability of spinal metastases. It classifies metastatic lesions into stable, unstable, and potentially unstable categories [11]. Studies on the SINS have rapidly increased in the last few years, and its usefulness and validity have been verified by some reports [15,16,17,18]. Numerous retrospective studies have reported the incidence of vertebral compression fracture and spinal cord compression after radiotherapy. Shi et al. reported that lesions categorized as unstable according to the SINS system are significantly more likely to develop into new or worsening vertebral fractures [16]. Additionally, a retrospective cohort study of 78 patients with single spinal metastasis following radiotherapy stated that an increased SINS was associated with spinal cord compression and vertebral compression fracture [18].

Furthermore, the SINS is used to predict skeletal-related events. According to a retrospective study of 47 patients with non-small cell lung cancer [19], patients with unstable or potentially unstable lesions were nearly 4 times more likely to experience a skeletal-related event than those with stable lesions. In addition, a multi-institutional retrospective series of 1509 patients with spinal metastases found that the mean SINS in operative patients (mean: 10.7) was significantly higher than in non-operative patients (mean 7.2) [20]. Pennington et al. analyzed 436 lesions in 51 patients with spinal metastases and reported that patients with a SINS ≥ 10 were more likely to require stabilization surgery than those with a SINS ≤ 9 [21]. 

In the original paper, Fisher et al. recommended that patients with lesions in the potentially unstable category (7–12) should be considered for spinal surgery [11]. However, few reports have statistically supported this recommendation. The spine surgeon and the author felt that a more concrete threshold is needed to decide on the use of surgery for spinal metastasis. From an analysis of 299 case series, patients with a SINS ≥ 11 following radiotherapy had a >2.5-fold increased risk of experiencing a spinal adverse event than those with a SINS ≤ 10 [22]. The true threshold of the SINS for SSM onset remains unclear. In the current study, the cut-off value of 9.5 was calculated via statistical analysis. Patients with a SINS ≥ 10 had a higher risk of SSM onset, and these patients developed SSM with a 62.5% probability in a median of 5.5 months. This result is particularly crucial to the management of spinal metastasis.

Regarding the treatment options for spinal metastasis patients with a SINS ≥ 10, previous reports demonstrated the effectiveness of radiation therapy in relieving pain caused by spinal metastasis [7,8]. As a result, radiation therapy is the first choice for spinal metastasis treatment. However, the short-term protective effect of this treatment on pathological fractures has not yet been established, and radiation therapy was not identified as a significant protective factor in this study. Furthermore, the use of BMAs was also not identified as a protective factor; however, other previous studies reported that BMA therapy strongly prevents skeletal-related events [23]. Consequently, BMA therapy should be considered for patients with a SINS ≥ 10. 

Single BMA therapy should be administered to patients who received continuous treatment for primary cancer, such as chemotherapy and molecularly targeted drugs. Compared with BMA therapy, surgery immediately improves PS, ADL, and neurological status with a 90% probability 1 month after surgery, and the clinical improvement is maintained for at least 6 months [5,24]. Needless to say, the surgical indication should be determined by comprehensively considering the patient’s prognosis and wish. The aim of treatment for metastatic disease is not only longer survival, but also better QOL. Obtaining a fine balance between survival and QOL is the ultimate goal of treatment for metastasis. If patients with a SINS ≥ 10 are predicted to survive for longer than 6 months, interventions consisting of percutaneous vertebroplasty, balloon kyphoplasty, and spinal surgery (stabilization +/− decompression) should be considered in addition to BMAs to prevent the onset of SSM. Furthermore, patients with a SINS ≥ 13 developed SSM with 100% probability within a mean of 1.37 months. Evidently, these patients must be considered for interventions, even if the chief complaint was mild at the first consultation. Thus, the authors recommend a prophylactic strategy for spinal metastasis according to the individual prediction of SSM onset and survival. 

This study has several limitations. Specifically, the overall rate of SSM onset (28.9%) appears to be relatively high. Selection bias may have been the cause as the patients in this study were referred to the orthopedic department by oncologists, radiotherapists, and others. Asymptomatic spinal metastases are difficult to define, especially in determining the pain component of the SINS. In this study, some patients represented three of pain component at study enrollment. However, these patients were considered asymptomatic because taking small quantities of pain killer did not interfere with ADL. In this way, the authors consider that this selection bias is appropriate for orthopedic clinical medicine. Another limitation is the recent progress in cancer treatment. Our study had no consideration of immune checkpoint inhibitors or new effective radiotherapies, including intensity-modulated radiotherapy and stereotactic body radiotherapy. These treatment options may be effective against the onset of SSM; therefore, further investigations are required in the future.

## 5. Conclusions

This study revealed the natural history of patients with spinal metastases and suggested that a SINS ≥ 10 is a significant risk factor for the onset of SSM. The aim of treatment for metastatic disease is not only longer survival, but also better QOL. Obtaining a fine balance between survival and QOL is the ultimate goal of treatment for metastasis. If patients with a SINS ≥ 10 are predicted to survive for longer than 6 months, the authors recommend a prophylactic strategy for spinal metastasis according to the individual prediction of SSM onset and survival. 

## Figures and Tables

**Figure 1 cancers-15-01251-f001:**
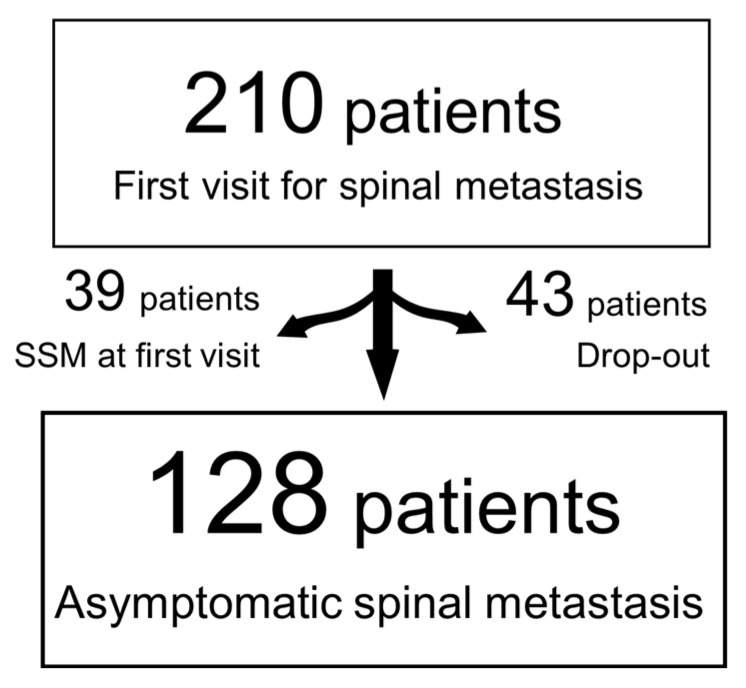
Flowchart of the study design.

**Figure 2 cancers-15-01251-f002:**
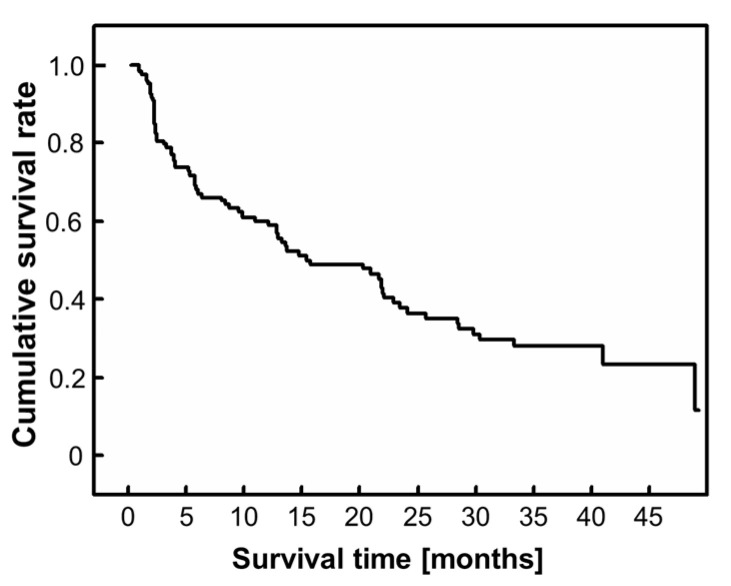
Kaplan–Meier survival curve.

**Figure 3 cancers-15-01251-f003:**
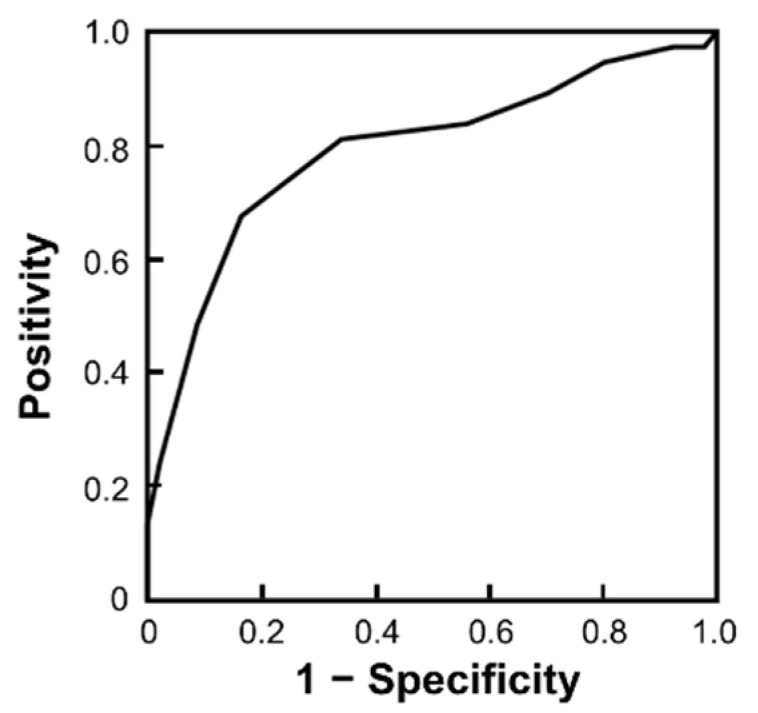
The receiver operating characteristic curve for the SINS. SINS: Spinal Instability Neoplastic Score.

**Figure 4 cancers-15-01251-f004:**
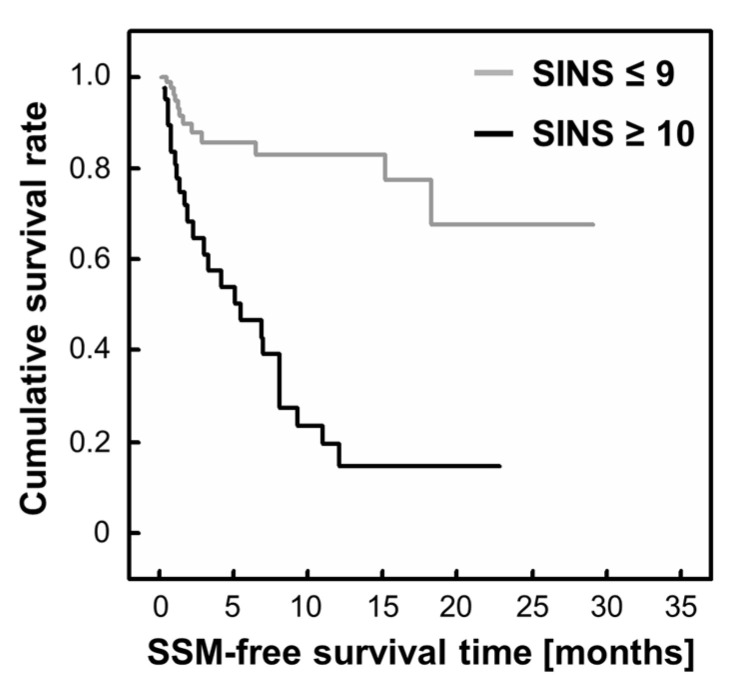
Kaplan–Meier SSM-free survival time curve according to the SINS (≥10 and ≤9). SINS: Spinal Instability Neoplastic Score; SSM: symptomatic spinal metastasis. *n* (SINS ≤ 9) = 88, *n* (SINS ≥ 10) = 40.

**Figure 5 cancers-15-01251-f005:**
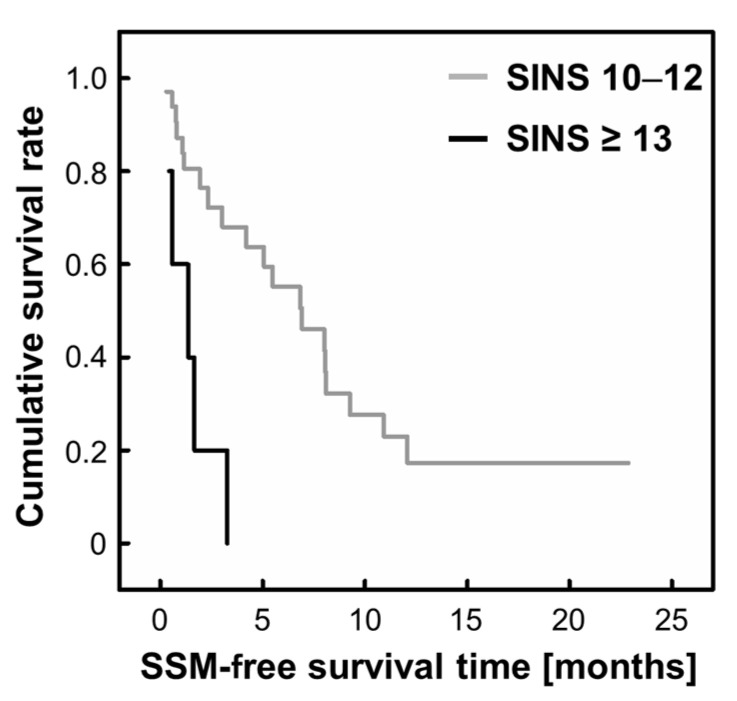
Kaplan–Meier SSM-free survival time curve according to the SINS (≥13 and 10–12). SINS: Spinal Instability Neoplastic Score; SSM: symptomatic spinal metastasis. *n* (SINS 10–12) = 35, *n* (SINS ≥ 13) = 5.

**Table 1 cancers-15-01251-t001:** Demographic and clinical factors at study enrollment.

Variables		Variables	
Age (no. (%))		Barthel index (mean [range]) (pts)	79.7 [5–100]
≥65	85 (66.4)	SINS (total score) (median [interquartile range]) (pts)	8 [7–10]
˂65	43 (33.6)	Location	2 [2–3]
Sex (no. (%))		Pain	1 [1–3]
Male	75 (58.6)	Bone lesion	2 [1–2]
Female	53 (41.4)	Alignment	0 [0–0]
Malignancy of primary site (no. (%))		Collapses	1 [1–2]
Lung, osteosarcoma, stomach, bladder, esophagus, pancreas	40 (31.3)	Spinal element	1 [0.25–3]
Liver, gallbladder, unidentified	18 (14.1)	Radiotherapy (no. (%))	
Others	26 (20.3)	Yes	90 (70.3)
Kidney, uterus	19 (14.8)	No	38 (29.7)
Rectum	4 (3.1)	Chemotherapy (no. [%])	
Thyroid, breast, prostate, carcinoid tumor	21 (16.4)	Yes	31 (24.2)
ECOGPS grade (no. (%))		No	97 (75.8)
PS 0	14 (10.9)	Molecularly targeted drugs or hormonal therapy (no. (%))	
PS 1	45 (35.2)	Yes	43 (33.6)
PS 2	35 (27.3)	No	85 (66.4)
PS 3	23 (18.0)	Bone-modifying agents (no. (%))	
PS 4	11 (8.6)	Yes	73 (57.0)
Median PS	2	No	55 (43.0)

ECOG: Eastern Cooperative Oncology Group; PS: performance status; SINS: Spinal Instability Neoplastic Score.

**Table 2 cancers-15-01251-t002:** Primary cancer types.

Primary Tumor	No. (%) of Patients
Lung	33 (25.8)
Kidney	15 (11.7)
Breast	10 (7.8)
Liver	10 (7.8)
Unknown	9 (7.0)
Thyroid	6 (4.7)
Prostate	5 (3.9)
Lymphoma	5 (3.9)
Myeloma	4 (3.1)
Bladder	4 (3.1)
Colorectal	4 (3.1)
Others	23 (23.8)

**Table 3 cancers-15-01251-t003:** Multivariate analysis of risk factors for symptomatic spinal metastases.

Variables	Odds Ratio	95% CI	*p*
Age (≥65)	1.062	0.399–2.830	0.904
Sex (male)	1.076	0.391–2.965	0.887
Malignancy of primary tumor	1.007	0.749–1.353	0.963
ECOGPS	1.419	0.645–3.126	0.384
Barthel Index	1.024	0.991–1.059	0.159
SINS (total score)	1.739	1.345–2.250	<0.001 *
Radiotherapy	2.413	0.745–7.811	0.142
Chemotherapy	1.603	0.478–5.369	0.444
Molecularly targeted drugs or hormonal therapy	1.012	0.330–3.100	0.984
Bone-modifying agents	0.535	0.200–1.431	0.213

ECOG: Eastern Cooperative Oncology Group; PS: performance status; SINS: Spinal Instability Neoplastic Score; CI: confidence interval; * *p* < 0.05.

## Data Availability

All data can be found in the text.

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
