# Peer review of "The Identification of Risk Factors for Symptomatic Spinal Metastasis Onset: A Prospective Cohort Study of 128 Asymptomatic Spinal Metastasis Patients"

_cancers, 2023, doi:10.3390/cancers15041251_

Round 1
Reviewer 1 Report
Thank you for the opportunity to read this interesting manuscript.
Kenichiro Kakutani and colleagues present a prospective study on identification of risk factors for symptomatic spinal metastasis onsets. The study comprises 128 patients with asymptomatic spinal metastases at inclusion and who were then followed-up. The study addresses a relevant clinical problem. Overall, the paper is well written and thoughtful, and provides important baseline data which may be useful for comparison in future studies.
I have some comments:
1. My major concern is about multiple regression analysis (Table 3). You analysed 16 different factors in 128 patients. I am not statistician, but I am worried that there is risk for overfitting. I think that the editor should consider statistical consultation.
2. You included patients between 2016 and 2018. When did you finish the study? How many patients were alive at the end of the study and what was the median follow-up for these patients.
3. Mechanical pain is one of the components included in calculation of SINS. In the Table 1, the range of pain component is 1-3, where 3 represents presence of mechanical pain. But this is still group with asymptomatic spinal metastases. This should at least be discussed.
4. What do you mean with history of radiotherapy at inclusion? Radiotherapy of primary tumour, radiotherapy of other metastases or radiotherapy of other bone metastases? This should be clarified.
5. How many of the asymptomatic patients at first consultation (inclusion) had the spinal bone metastasis as the first manifestation of previously unknown malignancy? These patients have usually better prognosis due to previously untreated malignant disease.
6. You defined SSM as spinal metastases associated with progressive neurological deficits or intractable pain resistant to conservative care. How many patients in the SSM group developed symptomatic spinal cord compression?
7. In figures 4 and 5, number of patients in each group should be presented.
Author Response
Thank you for the opportunity to read this interesting manuscript.
Kenichiro Kakutani and colleagues present a prospective study on identification of risk factors for symptomatic spinal metastasis onsets. The study comprises 128 patients with asymptomatic spinal metastases at inclusion and who were then followed-up. The study addresses a relevant clinical problem. Overall, the paper is well written and thoughtful, and provides important baseline data which may be useful for comparison in future studies.
I have some comments:
- My major concern is about multiple regression analysis (Table 3). You analysed 16 different factors in 128 patients. I am not statistician, but I am worried that there is risk for overfitting. I think that the editor should consider statistical consultation.
Answer; first of all, we would like to thank the reviewers for the insightful comments and suggestions. We really appreciate your comment. In response to your comment, I re-analyzed our data. As the results, BMAs and location component of SINS lost the significance. Based on these new results, we revised the manuscripts and table.
- You included patients between 2016 and 2018. When did you finish the study? How many patients were alive at the end of the study and what was the median follow-up for these patients.
Answer; thank you for good comment, we finished this study at 2020. Consequently we have no data regarding patient survival at this time. On the other hand, the median follow-up time was described in line 148-149.
- Mechanical pain is one of the components included in calculation of SINS. In the Table 1, the range of pain component is 1-3, where 3 represents presence of mechanical pain. But this is still group with asymptomatic spinal metastases. This should at least be discussed.
Answer; thank you for insightful comments and suggestions, we really appreciate your comment. Asymptomatic spinal metastases are difficult to define, especially to determine the pain component of SINS. In this study, some patients represent 3 of pain component at the study enrollment. However, these patients were considered asymptomatic because taking only some pain killers did not interfere with ADL. This kind of selection bias is still remained. We described this comments in discussion in line 260-263.
- What do you mean with history of radiotherapy at inclusion? Radiotherapy of primary tumour, radiotherapy of other metastases or radiotherapy of other bone metastases? This should be clarified.
Answer; thank you for your good comments. In this study, the history of radiation therapy is defined radiation therapy to metastatic spine that is the subject of this study. We revised this sentence in line 111-112.
- How many of the asymptomatic patients at first consultation (inclusion) had the spinal bone metastasis as the first manifestation of previously unknown malignancy? These patients have usually better prognosis due to previously untreated malignant disease.
Answer; thank you for your good comments. We completely agree with you. The patients with malignant tumors firstly discovered due to bone metastasis usually have a good prognosis. These patients may affect the results of this study. However, unfortunately we have no data before first consultation. We would like to obtain for future study.
- You defined SSM as spinal metastases associated with progressive neurological deficits or intractable pain resistant to conservative care. How many patients in the SSM group developed symptomatic spinal cord compression?
Answer; thank you for insightful comments and suggestions, we really appreciate your comment. In response to your comment, we re-analyzed our data. As the results, of 37 SSM patients, the cause of SSM were neurological deficit due to spinal cord compression in 15 cases and intractable pian due to pathological fracture in 22 cases. We described this sentence in line 156-158.
- In figures 4 and 5, number of patients in each group should be presented.
Answer; thank you for your good comments. We described the number of patients in each group in figure 4 and 5.
Reviewer 2 Report
The authors have presented a prospective cohort study that aims to identify the risks factors for symptomatic spinal metastasis onset. Your article is eloquent, methodic, and well written. Please find below some comments.
1. In the abstract, please add the timeline of your study (from 2016 to ?)
2. In lines 90-91 you have explained how spinal metastasis were diagnosed. Did you use all the technique mentioned or you choose one instead of another in certain situations (i.e. bone scintigraphy, positron emission tomography, needle biopsy)? Please clarify this point.
3. How do you explain the development of SSM in patients with a SINS < 9?
4. The discussion should be implemented. Please, explain better and extensively what your study would like to enhance and how it should change future studies.
5. You cannot write in the conclusion the word “partially”, it makes your work weak.
Author Response
The authors have presented a prospective cohort study that aims to identify the risks factors for symptomatic spinal metastasis onset. Your article is eloquent, methodic, and well written. Please find below some comments.
- In the abstract, please add the timeline of your study (from 2016 to ?)
Answer; first of all, we would like to thank the reviewers for the insightful comments and suggestions. We really appreciate your comment. We described the timeline in the abstract in line 28.
- In lines 90-91 you have explained how spinal metastasis were diagnosed. Did you use all the technique mentioned or you choose one instead of another in certain situations (i.e. bone scintigraphy, positron emission tomography, needle biopsy)? Please clarify this point.
Answer; thank you for your good comments. We basically diagnosed the spine metastasis by plain radiography, computed tomography, magnetic resonance imaging. However, when the diagnosis was hard, bone scintigraphy, positron emission tomography, and histological evaluation of needle biopsy samples were performed. In response to your comments, we revised the manuscript in line 91-94.
- How do you explain the development of SSM in patients with a SINS < 9?
Answer; thank you for insightful comments and suggestions, we really appreciate your comment. This question is quite important of this study. The patients with a SINS<9 developed SSM due to neurological deficit in 3 cases and intractable pian due to pathological fracture in 9 cases. This sentence was described in line 172-174.
- The discussion should be implemented. Please, explain better and extensively what your study would like to enhance and how it should change future studies.
Answer; thank you for insightful comments and suggestions. In response to your comments. We revised the manuscript.
- You cannot write in the conclusion the word “partially”, it makes your work weak.
Answer; thank you for insightful comments and suggestions. In response to your comments. We revised the manuscript.
Round 2
Reviewer 1 Report
The authors made suggested revision which further improved the manuscript.